# Dynamics of anti-*Strongyloides* IgG antibody responses and implications for strongyloidiasis surveillance in rural Amazonians: A population-based panel data analysis

**Fabiana M. de Paula**[1,2]*, **Bruna B. Gomes**[1,2], **Dirce Mary C. L. Meisel**[1,2], **Mônica da-Silva Nunes**[3,4], **Carlos E. Cavasini**[5], **Kézia K. G. Scopel**[6], **Ronaldo C. B. Gryschek**[1,2], **Marcelo U. Ferreira** [iD][4,7]*

1 Laboratório de Investigação Médica LIM-06, Hospital das Clínicas da Faculdade de Medicina, Universidade de São Paulo, São Paulo, São Paulo, Brazil, 2 Instituto de Medicina Tropical de São Paulo, Faculdade de Medicina, Universidade de São Paulo, São Paulo, São Paulo, Brazil, 3 Department of Medicine, Federal University of São Carlos, São Carlos, São Paulo, Brazil, 4 Department of Parasitology, Institute of Biomedical Sciences, University of São Paulo, São Paulo, São Paulo, Brazil, 5 Department of Dermatological, Infectious and Parasitic Diseases, Faculty of Medicine of São José do Rio Preto, São José do Rio Preto, São Paulo, Brazil, 6 Department of Parasitology, Microbiology and Immunology, Federal University of Juiz de Fora, Juiz de Fora, Minas Gerais, Brazil, 7 Global Health and Tropical Medicine (GHTM), Associate Laboratory in Translation and Innovation Towards Global Health (LA-REAL), Institute of Hygiene and Tropical Medicine, NOVA University of Lisbon, Lisbon, Portugal

* fabiana.paula@hc.fm.usp.br (FMP); muferrei@usp.br (MUF)

## Abstract

### Background

Human strongyloidiasis was recently incorporated into the World Health Organization roadmap for neglected tropical diseases targeted for control in 2021–2030. However, the prevalence, incidence, and clinical burden of *Strongyloides stercoralis* infection remain understudied in remote communities across the Amazon due to its chronic nature, usually with absent or unspecific clinical manifestations, and the lack of practical and sensitive diagnostics for large-scale use. Here, we apply repeated antibody testing to estimate the prevalence of anti-*Strongyloides* IgG responses and identify incident infections in five farming settlements in the Amazonas State of Brazil.

### Methodology/Principal findings

We used an in-house enzyme immunoassay, with a *S. venezuelensis* larval extract as the solid-phase antigen, to detect specific IgG antibodies in 898 plasma samples collected during consecutive cross-sectional surveys over 4 years from 426 study participants aged >3 months, with an average of 35.9 years. Overall, 465 (51.8%) samples tested positive. However, only two infections that had been detected by fecal microscopy at survey 1 (March-May 2010) were treated with ivermectin. Antibody prevalence rose from 45.9% in 2010 to 61.1% in 2013, consistent with an increased (re)exposure to infective larvae over

**Data availability statement:** All relevant data are within the paper and its Supporting Information files.

**Funding:** This research was supported by the National Institute of Allergy and Infectious Diseases, National Institutes of Health, United States of America (NIAID, https://www.niaid.nih.gov/) (U19 AI089681 to MUF), the Fundação de Amparo à Pesquisa do Estado de São Paulo, Brazil (FAPESP, https://fapesp.br/en) (2009/52729-9 to MUF, 2013/04236-9 to RCBG, and 2022/02401-1 to FMP), and the Fundação para a Ciência e Tecnologia, Portugal (FCT; https://www.fct.pt/en/), institutional projects UID/04413/2020 and LA-REAL LA/P/0117/2020 to MUF. FMP, KKGS, and MUF receive senior researcher scholarships from the Conselho Nacional de Desenvolvimento Científico e Tecnológico of Brazil (CNPq, https://www.gov.br/cnpq/pt-br). The funders had no role in study design, data collection and interpretation, or the decision to submit the work for publication.

**Competing interests:** The authors have declared that no competing interests exist.

time. On average, there were 24.5 seroconversion events (a proxy of recent exposure to infection) per 100 person-years of follow-up, with 18.1 seroreversion events per 100 person-years. Nearly all participants with high antibody levels (i.e., above the median absorbance of seropositive tests) remained seropositive over the next years, with a single instance of high-to-nil antibody transition. Long-lasting high-level IgG responses were most likely due to frequent re-exposure to infective *S. stercoralis* larvae, chronic carriage of adult worms in the absence of treatment, or both. Conversely, over one-third of participants with low anti-*Strongyloides* antibody levels had transient IgG responses and seroreversed within 12 months.

## Conclusions/Significance

The results support the use of repeated antibody testing for monitoring temporal changes in *S. stercoralis* transmission in remote populations.

### Authors' summary

The epidemiology of strongyloidiasis – a neglected infection caused by the soil-transmitted nematode, *Strongyloides stercoralis* – remains largely understudied in remote tropical communities due to the lack of practical and sensitive diagnostic tests. Serology has been increasingly used for diagnosis at the individual and population levels, but few longitudinal antibody datasets are currently available. Here, we characterize the dynamics of anti-*Strongyloides* IgG antibody response among 426 residents in farming settlements in the Amazon Basin of Brazil. The overall seropositivity rate was 51.8% and rose from 45.9% in 2010 to 61.1% in 2013, with an average of 24.5 seroconversion events per 100 person-years of follow-up, consistent with an intense exposure to new infections over four years. Specific IgG responses persisted for at least 12 months in most seropositive study participants, especially among those with high baseline antibody levels, but one-third of low-level antibody responses were transient in the absence of specific anthelminthic treatment with ivermectin. The overall seroreversion rate was 18.1 events per 100 person-years of follow-up. We suggest that repeated antibody testing can be useful for identifying new *S. stercoralis* infections and monitoring epidemiological trends of human strongyloidiasis in remote populations.

## Introduction

Human strongyloidiasis, a common yet understudied soil-transmitted helminth (STH) infection caused by the nematode *Strongyloides stercoralis* [1], has recently been included in the World Health Organization roadmap for neglected tropical diseases targeted for control in 2021−2030 [2]. There are an estimated 600 million people infected worldwide, distributed mainly in low-income tropical countries and in the most disadvantaged communities of middle-income countries [3].

The prevalence, incidence, and clinical burden of *S. stercoralis* infection remain difficult to ascertain at the local, regional and global levels due to its chronic nature, with absent or unspecific clinical manifestations in most immunocompetent hosts, along with the lack of practical and sensitive fecal-based diagnostics appropriate for large-scale use [1]. The definitive diagnosis usually relies on the detection of *S. stercoralis* larvae in the stool, but there has

been no fecal-based diagnostic method widely accepted as a gold standard [4]. Direct fecal smear examination is straightforward but poorly sensitive, especially in chronic infections with low larval output [5]. The Baermann concentration method and the Koga agar plate culture are both more sensitive than direct fecal smear examination, but they require fresh stool samples and are time-consuming and cumbersome for use in population screening [ref. 5; but see also ref. 6]. Multiple stool specimens collected over consecutive days should ideally be processed and examined because of the intermittent release of larvae [7], but this is rarely feasible in community-wide surveys. Conventional and real-time polymerase chain reaction (PCR) protocols have a diagnostic accuracy comparable to that of the Baermann method and agar plate culture in clinical settings [8,9], but they are technically demanding and costly. More sensitive and practical diagnostics for use in resource-poor laboratories in endemic countries are needed to identify and treat cases, eliminate potential sources of infection, and accelerate progress toward the control of strongyloidiasis through population-based surveillance [1,5].

Antibody-based tests for strongyloidiasis have an estimated diagnostic sensitivity ranging from 71% to 95% [10,11]. They have been increasingly used for diagnosis at the individual and population levels, but the specificity can be low in tropical settings due to persisting IgG responses caused by past *S. stercoralis* infection and cross-reactivity with other locally prevalent STHs [5,12]. Changes in antibody status and levels over time can help identifying incident infections, most of them subclinical, that are often missed by surveillance relying on suboptimal stool microscopy [5]. Moreover, antibody titers usually decrease after anthelminthic therapy in patients who are not exposed to reinfection, and antibodies tend to become undetectable within one year [13]. However, seroconversion patterns and the duration of specific antibody responses have been little studied in endemic settings in the presence or absence of specific treatment. Limited data show significantly decreased antibody positivity rates 6-12 months after the mass administration of 200 μg/kg of ivermectin [14,15], but IgG responses can persist in up to 25% of treated individuals, mainly among those with high baseline antibody titers [16].

Here, we use repeated antibody measurements to estimate the prevalence of anti-*Strongyloides* IgG responses over four years and identify incident infections in remote farming settlements intermingled with the rainforest in the Amazon Basin of Brazil. We discuss the use and interpretation of antibody tests for strongyloidiasis surveillance in endemic settings.

## Methods

### Ethics statement

The study was carried out according to the principles of the Declaration of Helsinki. Study protocols were approved in early 2010 by the Institutional Review Board of the University Hospital of the University of São Paulo (1025/10) and by the National Human Research Ethics Committee of the Ministry of Health of Brazil (551/2010). The ethical clearance has been renewed annually by the Institutional Review Board of the University Hospital of the University of São Paulo. Written informed consent was obtained from all study participants >18 years of age or from parents or caregivers of children. Participants aged 6-18 years also were requested to sign an assent form; those unable to sign their name provided a thumb print. Plasma samples tested for anti-*Strongyloides* antibodies were part of the biological collection, maintained at the Institute of Biomedical Sciences, University of São Paulo, that resulted from 10 consecutive cross-sectional surveys carried out in the study site. The use of experimental animals (Wistar rats) for antigen preparation was approved by the Animal Research Ethics Committee of the Faculty of Medicine of the University of São Paulo (0356A).

## Study site

We measured anti-*Strongyloides* IgG antibodies in remote populations from southern Amazonas state, Western Brazilian Amazon, next to the border with Bolivia (S1 Fig). The study site is characterized by an equatorial humid climate (annual average temperature, 26.4°C), with most rainfall between November and March (annual average, 2,318 mm). There are five small farming settlements in the area, where the main economic activities are subsistence agriculture and logging in the nearby rainforest [17]. Newly opened informal farming settlements such as our study site, characterized by poor housing and lack of basic infrastructure, are commonly seen across the "deforestation arc" of the Amazon, especially along or next to major roads such as the BR-364 and BR-319 interstate highways. Tropical diseases such as malaria and STH infections are expected to be highly prevalent in these settings.

The main settlement, Remansinho, is situated along the final 40 km of a 60 km-long unpaved road originating from the BR-364 interstate highway, known as Ramal do Remansinho, while the other four are situated along secondary roads known as Ramal da Linha 1, Ramal da Castanheira, Ramal dos Seringueiros, and Ramal dos Goianos, all originating from the main unpaved road. Ramal da Linha 1 and Ramal da Castanheira are older settlements, opened in the late 1990s, whereas the colonization of the other areas started only in 2007. There is no electricity or piped water supply in the area. Health care access is very limited in the Remansinho area. There is no formal health facility in the community, except for a malaria diagnosis post with a microscopist. Patients seek basic health care in the nearest village, Nova Califórnia (40-60 km away by a dirt road), and more specialized care in a regional hospital in the village of Extrema, >80 km away.

## Study design and population

A prospective panel study was initiated in March 2010 to investigate the epidemiology of malaria and other tropical infectious diseases in the local population [17]. The present analysis comprises biological samples and data from 10 consecutive cross-sectional surveys of the entire population of the five settlements: survey 1, March-May 2010; survey 2, May-July 2010; survey 3, October-November 2010; survey 4, March-April 2011; survey 5, October-November 2011; survey 6, April-May 2012; survey 7, October-November 2012; survey 8, April-May 2013; survey 9, October-November 2013; and survey 10, September-October 2014. A total of 584 residents were enumerated during the consecutive cross-sectional surveys (1 through 10) between 2010 and 2014. Sociodemographic and morbidity information and venous blood samples were collected from consenting residents [17]. Information on ownership of selected household assets was combined to derive a wealth index [18], which was stratified into quartiles for statistical analysis.

The study population comprises 426 study participants ranging from 4 months to 79 years of age (mean, 35.9 years; standard deviation, 0.61), distributed into 160 households, who had one or more plasma samples available for antibody testing (72.9% of those enumerated during our census surveys). Venous blood samples were drawn into heparinized vacuum tubes during house-to-house visits carried out from 8 am to 2 pm between 2010 and 2014. Samples were kept on ice packs until centrifugation for plasma separation, within up to 8 hours, in our field laboratory in Acrelândia, Acre, situated 120 km west of the Remansinho area (S1 Fig). Plasma aliquots were kept in Acrelândia at -20°C for up to four weeks and shipped on dry ice to our research laboratory in São Paulo, where they were stored at -70°C until tested. Samples were screened for anti-*Strongyloides* IgG antibodies in 2022, with a total of 898 plasma samples analyzed. Between 1 to 5 plasma samples per participant were analyzed. Because of the high mobility of the local population, only 5.4% (n = 23) of the study participants were present in

the site and had specific antibodies measured in all study years; 180 (42.2%) were tested for antibodies only once, 116 (27.2%) were tested twice, 57 (13.4%) were tested three times, and 50 (11.7%) were tested four times. For participants who provided >1 sample throughout the study, a single sample collected each year – in 2010 (survey 1, 2, or 3), 2011 (survey 4 or 5), 2012 (survey 6 or 7), 2013 (survey 8 or 9), and/or 2014 (survey 10) – was tested for antibodies.

During survey 1, participants were given plastic containers containing 10% formalin (Coprotest, NL, Campinas, Brazil) and asked to provide one stool sample for detection of eggs, cysts, and larvae of intestinal parasites. We used a simple sedimentation-concentration technique in 100-mm conic-bottom test tubes [19,20] that appears to be nearly as sensitive for the diagnosis of strongyloidiasis as the Baermann and agar plate culture methods [20]. Overall, 152 participants who were tested for anti-*Strongyloides* antibodies in survey 1 had also stool samples collected and examined for intestinal parasites. Study participants with microscopically confirmed infections were offered free treatment with ivermectin (200 μg/kg/day for 2 days) for *S. stercoralis*, albendazole (400 mg, single dose) for other STHs, or metronidazole (adults: 500-750 mg three times daily over 7 days; children, 30-50 mg/kg/day) for pathogenic protozoa.

## Antigen preparation

We used a total extract of third-stage *S. venezuelensis* larvae, prepared as described [21], as the solid-phase antigen for antibody capture. Briefly, we obtained 200 mg of dried third-stage larvae harvested from charcoal cultures of feces from experimentally infected Wistar rats. Larvae were resuspended in phosphate-buffered saline (PBS) at pH 7.2 containing 0.25% of the cationic detergent cetyltrimethylammonium bromide (CTAB) and incubated overnight at 4°C to extract surface cuticle antigens. The suspension was centrifuged ($12,400 \times g$ for 10 minutes) and the supernatant was harvested and stored at -20°C until use.

## Antibody detection

We used an enzyme immunoassay (ELISA) for IgG detection, with an estimated diagnostic sensitivity of 95.0%, with a specificity of 97.8% [21]. High-binding 96-well microplates were incubated overnight at 4°C with 10 μg/mL of capture antigen diluted in carbonate/bicarbonate buffer (pH 9.6). After washing steps with PBS containing 0.1% Tween 20 (PBS-T), microplates were incubated with PBS-T plus 5% skim milk (PBS-TM). Plasma samples were incubated in duplicate, at 1:200 dilution in PBS-TM, for 45 minutes at 37°C. Next, we added peroxidase-conjugated, mouse anti-human IgG antibody (Sigma-Aldrich, St. Louis, MI, USA) diluted at 1:10,000 in PBS-TM, and microplates were incubated for 45 minutes at 37°C. We next added the 3,3', 5,5-tetramethylbenzidine (TMB) chromogen solution (Thermo Fischer Scientific, Waltham, MA, USA) for a 7-min incubation at room temperature, in the dark. Finally, the reaction was stopped with 2N $H_2SO_4$ and absorbance was measured at 450 nm. All microplates included positive controls (a pool of antibody-positive plasma samples from patients with *S. stercoralis* infection confirmed by detection of larvae with the Baermann concentration method) and negative controls (a pool of antibody-negative plasma samples from unexposed, apparently health people), as well as blank controls (no plasma added to microplate wells). We accepted up to 10% variation in absorbance readings from positive and negative control sera across microplates. Samples giving absorbance readings up 10% above or below the previously determined cut-off value of 0.286 [21] were retested to confirm their positive or negative status. Plasma samples with absorbance values higher than the cut-of value were classified as positive. Antibody-positive samples were further classified as "low antibody response" (absorbance between 0.287 and 0.561) and "high antibody response" (absorbance above the median value of 0.561, calculated for all positive samples tested across all surveys).

## Statistical analysis

Data were analyzed with Stata 18 (StataCorp, College Station, TX, USA). Standard descriptive statistics were used to summarize the main exposure and outcome variables. Statistical significance was defined at the 5% level; 95% confidence intervals (CI) of proportions and rates and interquartile ranges (IQR) were estimated when appropriate. Correlations between pairwise absorbance levels were evaluated using the nonparametric Spearman correlation ($r_s$) test.

Multivariable logistic regression models were run to identify correlates of anti-*Strongyloides* antibody response in the study population (main study outcome, yes/no) by combining results from one to five observations (antibody measurements and associated metadata) per participant. Unadjusted models comprised the following individual-level covariates: age (stratified as ≤ 5, 6-15, 16-30, 31-50, and >50 years); gender; whether the participant reported treatment with locally available anthelminthics (albendazole 400 mg, single dose; or mebendazole, 100 mg twice a day for 3 days) within the past 6 months (yes/no); whether the participant reported having passed helminths in the feces within the past 6 months, either spontaneously or following anthelminthic treatment (yes/no); and type of stool disposal (pit latrine vs. open defecation). Household-level covariates were wealth index quartile; presence of a groundwater well in the property (yes/no); housing material (bricks, wood, or other materials, such as adobe or palm leaves); and place of residence (farming settlement), in addition to survey year (2010, 2011, 2012, 2013 or 2014). Variables associated with $P < 0.20$ in initial unadjusted analyses were entered in multivariable regression models built for all surveys combined; participants with missing information (n = 44) were excluded from the final model. We used the Stata command "xtset" to set the panel variable (individual, *id*) and the time variable (*year*). We had repeated observations for most participants (up to five observations per individual; grouping variable, *id*), who were also clustered within households (grouping variable, *household*). Due to the nested structure of the data, we used the Stata command "melogit" to build mixed-effects logistic regression models that included the grouping variables as random factors. Odds ratio (OR) estimates are provided along with 95% CIs to quantify the influence of each predictor on the outcome (antibody positivity) while controlling for all other covariates.

The secondary study outcome was seroconversion within 12 months, which was taken as a proxy of recent exposure to infection. Seroconversion was defined as an IgG-positive test (absorbance above the cut-off value of 0.286) preceded by a negative test in the previous survey, 12 months earlier. We estimated the seroconversion rate as the number of events per 100 person-years at risk, and its Fisher's exact 95% CIs was calculated, with the time at risk defined as the time interval between two blood draws from the same participant. We also estimated the number of seroreversion events per 100 person-years at risk, by defining a seroreversion event as an IgG-negative test preceded by a positive test 12 months earlier.

## Results

### Antibody prevalence and its correlates

Anti-*Strongyloides* IgG antibodies were often detected in the study population, consistent with high exposure to infection. Overall, 465 of 898 (51.8%; 95% CI, 48.4-55.1%) plasma samples that were collected over nearly five years, from March 2010 to October 2014, from 426 study participants tested positive for IgG antibodies; 232 (25.8%, 23.0-28.8%) samples had antibody levels above the median (absorbance value > 0.561) or "high antibody levels". Positivity rates varied significantly across age groups ($\chi^2 = 9.605$, 4 degrees of freedom, $P = 0.048$), ranging from 44.6% (95% CI, 38.7-50.7%) among adolescents and young adults aged 16-30 years to 58.3% (95% CI, 27.7-84.8%) in under-five children, but there was no significant trend toward increased or decreased IgG positivity rate with increasing age ([Table 1]).

Antibody prevalence rates increased over the study period and ranged from 45.9% (95% CI, 39.5-52.4%) in 2010 to the peak of 61.1% (95% CI, 51.4-70.1%) in 2013. Indeed, survey year was the only covariate found to be significantly associated with seropositivity in multivariable logistic regression analysis (*P*-value for trend = 0.009; Table 1). Only 14 (9.2%; 95% CI, 5.1-15.0%) of 152 participants tested during survey 1 had helminth eggs or larvae detected by stool examination. Two of them passed *S. stercoralis* larvae (prevalence, 1.3%) and both were positive for specific IgG antibodies (S1 Table). Similar proportions of helminth-positive (7 of 14; 50.0%) and helminth-negative (65 of 138; 47.1%) participants had anti-*Strongyloides* antibodies at the baseline (Fisher exact test, *P* = 1.000). Periodic deworming was a common practice and a likely reason for the low prevalence of STH infection observed in the population. Nearly a quarter of the study participants – a total of 200 (24.3%) out of 822 observations; information missing for 76 interviews – reported having taken albendazole or mebendazole within the past six months, regardless of any symptom or laboratory diagnosis of infection. Participants reported having passed helminths (mostly roundworms) in the feces within the past six months during 97 (12.1%) of 800 interviews (information missing for 98; Table 1).

## Seroconversion and seroreversion rates

Changes in anti-*Strongyloides* antibody status over time are shown in Figs 1, S2, S3, S4 and S5 Tables. We first describe changes in antibody responses in paired samples collected 12 months apart (Fig 1A).

To estimate the seroconversion rate within one year, we analyzed 213 sample pairs from study participants with an antibody-negative test who were reassessed approximately 12 months later (Fig 1A), with a total of 216.3 person-years of follow-up. We observed 53 seroconversion events among 51 participants (two participants seroconverted twice): 20 events between 2010 and 2011 (74 sample pairs tested), 18 between 2011 and 2012 (70 pairs tested), 12 between 2012 and 2013 (48 pairs tested) and 3 between 2013 and 2014 (21 pairs tested). Out of 213 initially seronegative participants, 47 (22.1%) had low antibody titers detected 12 months later (nil-to-low conversions) and 6 (2.8%) had high antibody titers detected 12 months later (nil-to-high conversions), as shown in S2 Table (italicized numbers). The overall seroconversion rate, considering both nil-to-low and nil-to-high conversions, was 24.5 (95% CI, 18.3-32.0) events per 100 person-years of follow-up. Because antibody tests were carried out approximately 10 years after the field study was concluded, no attempt was made to administer ivermectin to seroconverters in this highly mobile population.

Individual antibody trajectories of the 51 seroconverters are shown in Fig 2. Median absorbance values increased modestly, from 0.233 (IQR, 0.191-0.260) 12 months prior to seroconversion to 0.355 (IQR, 0.326-0.428) at seroconversion (S6 Table). There were only 6 (11.3%) transitions from nil to high antibody levels (four in 2011, one in 2012 and one in 2013), with absorbance at seroconversion ranging from 0.598 to 1.195 among them (S2 Table). The rate of nil-to-high antibody transition was 2.8 (95% CI, 1.0 to 6.0) events per 100 person-years of follow-up.

We next analyzed 195 sample pairs from study participants with an initial antibody-positive test who were reassessed approximately 12 months later (Fig 1A and S2 Table), with a total of 198.9 person-years of follow-up. We observed 36 seroreversion events within 12 months: 6 between 2010 and 2011 (51 sample pairs tested), 21 between 2011 and 2012 (71 pairs tested), 6 between 2012 and 2013 (47 pairs tested) and 3 between 2013 and 2014 (26 pairs tested). The seroreversion rate was estimated at 18.1 (95% CI, 12.7-25.0) events per 100 person-years of follow-up. As expected, since antibody prevalence rates increased with time, seroconversions during the study were slightly more frequent than seroreversions, with a rate ratio of 1.35 (95% CI, 0.87-2.13). No seroreverser had received ivermectin treatment for

**Table 1. Factors associated with the presence of anti-*Strongyloides* IgG antibodies in the population of five farming settlements in Amazonas State, Brazil (2010-14) in unadjusted (empty model) and adjusted multiple logistic regression analysis.**

| | n/N (%) | Unadjusted analysis | | | Adjusted analysis | | |
|---|---|---|---|---|---|---|---|
| | | OR | 95% CI | *P* | OR | 95% CI | *P* |
| **Individual-level variables** | | | | | | | |
| **Age** | | | | | | | |
| ≤ 5 years | 7/12 (58.3) | 3.26 | 0.98-108.32 | 0.508 | 2.08 | 0.04-112.88 | 0.719 |
| 6-15 years | 68/124 (54.8) | 1.03 | 0.29-3.60 | 0.964 | 1.46 | 0.33-6.53 | 0.621 |
| 16-30 years | 124/278 (44.6) | 0.46 | 0.18-1.16 | 0.100 | 0.50 | 0.17-1.52 | 0.224 |
| 31-50 years | 143/249 (57.4) | 1.13 | 0.41-3.11 | 0.805 | 1.43 | 0.44-4.67 | 0.551 |
| > 50 years | 115/226 (50.9) | 1 | Reference | | | | |
| | | *P* for trend | | 0.584 | *P* for trend | | 0.956 |
| **Gender** | | | | | | | |
| Female | 179/377 (47.5) | 1 | Reference | | 1 | Reference | |
| Male | 286/521 (54.9) | 1.58 | 0.81-3.07 | 0.176 | 1.70 | 0.75-3.84 | 0.205 |
| **Self-reported use of anthelminthics within the past 6 months?** | | | | | | | |
| No | 318/622 (51.1) | 1 | Reference | | | | |
| Yes | 108/200 (54.0) | 1.09 | 0.52-2.31 | 0.814 | | | |
| **Self-reported helminths passed in the feces within the past 6 months?** | | | | | | | |
| No | 363/703 (51.6) | 1 | Reference | | | | |
| Yes | 53/97 9 (54.6) | 1.08 | 0.30-3.81 | 0.908 | | | |
| **Type of stool disposal** | | | | | | | |
| Pit latrine | 211/418 (50.5) | 1 | Reference | | 1 | Reference | |
| Open defecation | 224/426 (52.6) | 1.78 | 0.80-3.99 | 0.158 | 1.65 | 0.70-3.89 | 0.257 |
| **Household-level variables** | | | | | | | |
| **Wealth index quartile** | | | | | | | |
| 1 (poorest) | 78/127 (61.4) | 1 | Reference | | 1 | Reference | |
| 2 | 104/215 (48.4) | 0.63 | 0.19-2.07 | 0.446 | 0.77 | 0.22-2.65 | 0.683 |
| 3 | 150/264 (56.8) | 0.96 | 0.27-3.42 | 0.951 | 1.17 | 0.31-4.37 | 0.813 |
| 4 (least poor) | 98/224 (43.7) | 0.31 | 0.09-1.07 | 0.065 | 0.53 | 0.15-1.87 | 0.323 |
| | | *P* for trend | | 0.089 | *P* for trend | | 0.290 |
| **Presence of a groundwater well in the property?** | | | | | | | |
| No | 114/196 (58.2) | 1 | Reference | | | | |
| Yes | 327/656 (49.8) | 0.54 | 0.21-1.39 | 0.201 | | | |
| **Housing material** | | | | | | | |
| Bricks | 19/36 (52.8) | 1 | Reference | | | | |
| Wood | 407/792 (51.4) | 0.82 | 0.12-5.77 | 0.843 | | | |
| Other materials | 15/24 (62.5) | 2.80 | 0.09-86.49 | 0.556 | | | |
| **Farming settlement of residence** | | | | | | | |
| Remansinho | 298/580 (51.4) | 1 | Reference | | 1 | Reference | |
| Goianos | 20/38 (52.6) | 0.54 | 0.09-2.97 | 0.475 | 0.19 | 0.03-1.18 | 0.074 |
| Linha 1 | 46/96 (47.9) | 0.53 | 0.14-1.97 | 0.345 | 0.45 | 0.10-2.04 | 0.303 |
| Castanheira | 77/123 (62.6) | 1.48 | 0.49-4.47 | 0.488 | 1.52 | 0.44-5.24 | 0.504 |
| Seringueiros | 17/48 (35.4) | 0.35 | 0.09-1.29 | 0.116 | 0.33 | 0.06-1.91 | 0.214 |
| **Survey year** | | | | | | | |
| 2010 | 112/244 (45.9) | 1 | Reference | | 1 | Reference | |
| 2011 | 121/223 (54.3) | 2.57 | 1.46-4.53 | 0.001 | 2.72 | 1.53-4.85 | 0.001 |
| 2012 | 107/217 (49.3) | 2.01 | 0.98-4.13 | 0.056 | 2.02 | 0.98-4.18 | 0.056 |
| 2013 | 69/113 (61.1) | 4.13 | 1.80-9.45 | 0.001 | 3.68 | 1.54-8.75 | 0.003 |

*(Continued)*

**Table 1.** (Continued)

| | n/N (%) | Unadjusted analysis | | | Adjusted analysis | | |
|---|---|---|---|---|---|---|---|
| | | OR | 95% CI | *P* | OR | 95% CI | *P* |
| 2014 | 56/101 (55.4) | 3.14 | 1.43-6.93 | 0.004 | 3.11 | 1.34-7.18 | 0.008 |
| | | *P* for trend | | 0.004 | *P* for trend | | 0.009 |

OR = odds ratio.

CI = confidence interval.

n = number of positive samples with IgG antibodies detected by ELISA among participants within the exposure category.

N = total number of samples from participants within the exposure category that were tested for IgG antibodies by ELISA.

Percentages (n/N × 100) correspond to the seroprevalence within each exposure category.

The total number of participants distributed across exposure categories may vary for some variables due to missing data.

The adjusted logistic regression model included 814 observations from 382 participants distributed into 135 households; participants with missing information (n = 44) were excluded.

strongyloidiasis during our study, although many reported recent deworming with albendazole or mebendazole. Importantly, these anthelminthics were commonly used at doses with little, if any, efficacy against *S. stercoralis* [22,23]. One of the two participants given ivermectin following fecal-based diagnosis of *S. stercoralis* infection in survey 1 (2010) sustained a high specific IgG response over the next 3 years (no sample collected in 2014); the second *S. stercoralis*-positive participant was lost for follow-up. Importantly, all seroreversers had low IgG antibody levels at the baseline, 12 months earlier (Fig 1A and S2 Table).

## Long-term changes in antibody responses

To further examine whether high levels of IgG antibodies persisted over time, we extended our analysis to sample pairs collected from untreated individuals more than one year apart (Fig 1B, 1C, and 1D and S3, S4, and S5 Tables). Transitions from high to nil IgG antibody response were not observed in sample pairs collected either 36 months apart (n = 29 comparisons; Fig 1C and S4 Table) or 48 months apart (n = 6 comparisons; Fig 1D and S5 Table). The only instance of high-to-nil transition was found in only one out of 62 participants tested two years apart, who had a high antibody response in 2010 but was antibody-negative in 2012 (Fig 1B and S3 Table). Overall, untreated individuals with high IgG antibody levels at the baseline usually remained seropositive over the next years.

Conversely, study participants with low anti-*Strongyloides* IgG responses often tested negative after 12 months (36 of 100 comparisons, 36.0%; Fig 1A and S2 Table), after 24 months (19 of 62, 30.6%; Fig 1B and S3 Table), after 36 months (5 of 28, 17.9%; Fig 1C and S4 Table) and after 48 months (4 of 9, 44.4%; Fig 1D and S5 Table). Untreated individuals with low-level and transient IgG response to larval antigens may had been exposed to third-stage larvae that failed to develop into adult worms, may have cleared infections with adult *S. stercoralis* females spontaneously, or may present cross-reactive antibodies with other STHs. The levels of antibodies in the same participants were strongly correlated across years; Spearman correlation coefficients ($r_s$) ranged between 0.636 and 0.850, with *P* < 0.0001 for all pairwise comparisons (S7 Table).

## Discussion

Here we use repeated antibody testing to reveal incident and persistent *S. stercoralis* infections in remote farming settlements in the Amazon Basin of Brazil. Approximately 46% of the study participants had anti-*Strongyloides* IgG antibodies at the baseline in 2010 and nearly 25% of

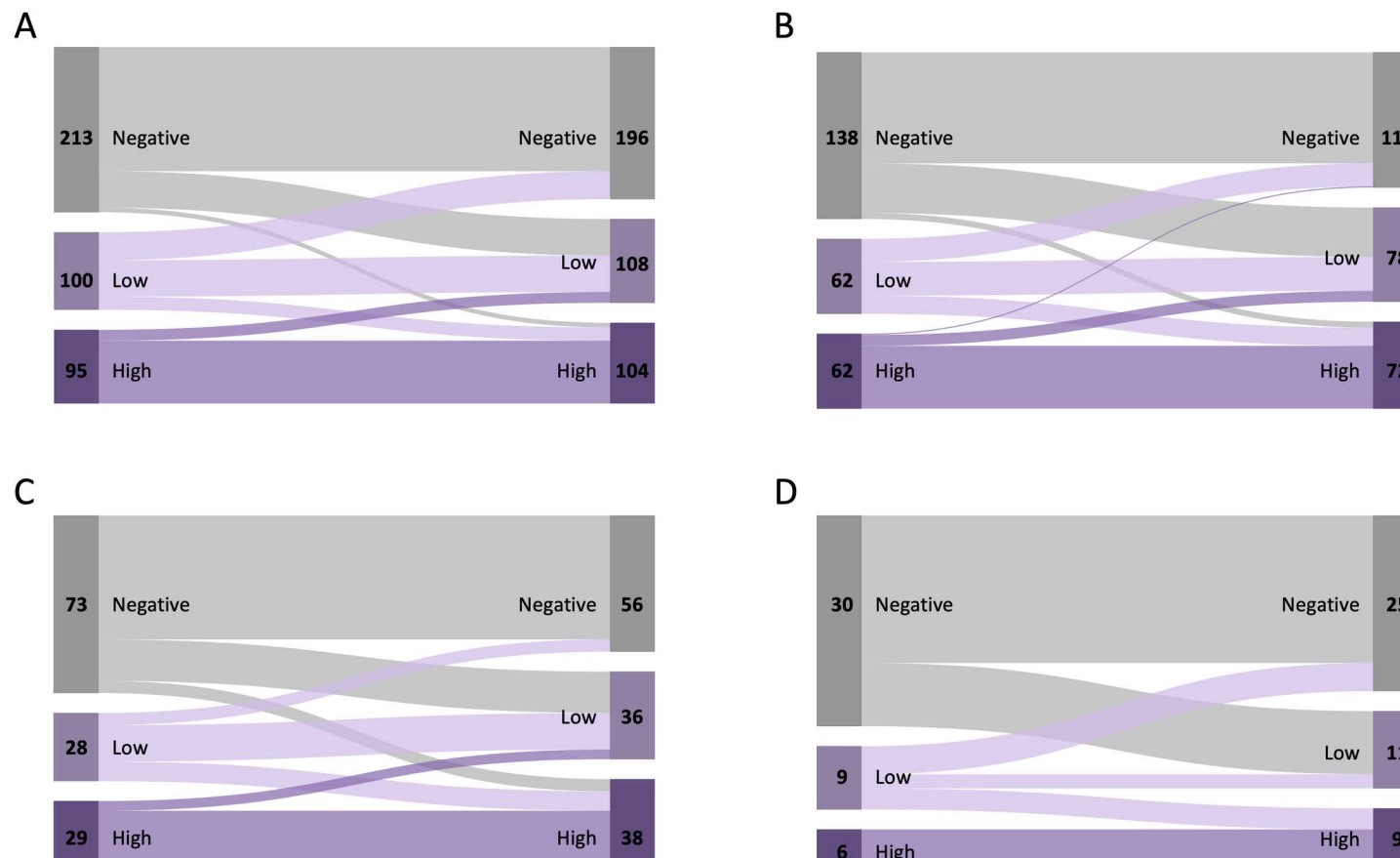

**Fig 1. Distribution of participants in consecutive surveys carried out in farming settlements in Amazonas State, Brazil, according to anti-*Strongyloides* IgG antibody status.** Panel A, blood draws approximately 12 months apart (data from 2010 vs. 2011, 2011 vs. 2012, 2012 vs 2013, and 2013 vs. 2014; **n** = 408 comparisons); Panel B, blood draws approximately 24 months apart (data from 2010 vs. 2012, 2011 vs. 2013, and 2012 vs. 2014; **n** = 262 comparisons); Panel C, blood draws approximately 36 months apart (data from 2010 vs. 2013 and 2011 vs. 2014; **n** = 130 comparisons); Panel D, blood draws approximately 48 months apart (2010 vs. 2014; **n** = 45 comparisons). Anti-*Strongyloides* IgG responses were stratified as negative (absorbance ≤ 0.286; light gray), low (absorbance between 0.287 and 0.561; light purple) and high (absorbance > 0.561; purple), with the absorbance value of 0.561 corresponding to the median absorbance value among positive samples during the study. Numbers of individuals within each antibody status category are shown. Data from Panel A (see also the italicized numbers in S2 Table) were used to estimate the 12-month seroconversion rate: of 213 initially seronegative participants, 47 (22.1%) had low antibody titers detected 12 months later (nil-to-low conversions) and 6 (2.8%) had high antibody titers detected 12 months later (nil-to-high conversions).

the initially antibody-negative individuals seroconverted per year of follow-up, consistent with frequent (re)exposure to infective larvae.

Population-wide screening of strongyloidiasis should ideally use biological samples that are easy to obtain, transport and store, and diagnostic methods that require simple laboratory equipment and only basic personnel skills. Indeed, the lack of sensitive, practical, and field-deployable diagnostics has severely limited our understanding of the epidemiology and burden of strongyloidiasis in rural Amazonians, but serology can help to fill this knowledge gap.

Estimates of the countrywide prevalence of *S. stercoralis* infection in Brazil, based on stool examination, range between 5.5% [24] and 11.2% [4]. Those for Amazonas State, the only state in the Amazon Basin of Brazil with available prevalence data, vary widely between 0.8% and 14.1% [24]. Seroprevalence rates as high as 72.6% have been reported in Amazonian communities in Peru [25] and 22.7% in Ecuador [26]. The proportions of

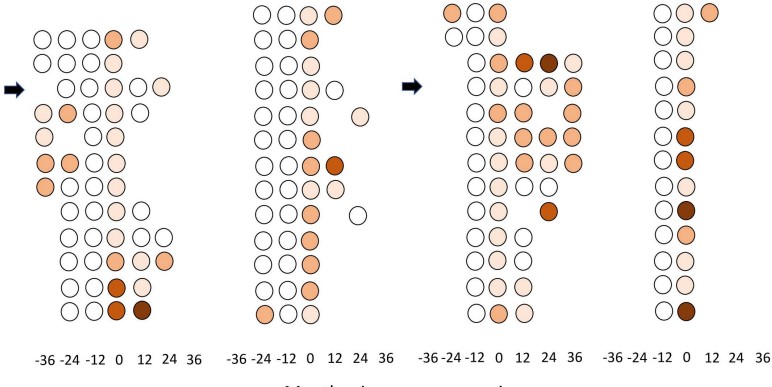

-36 -24 -12  0  12 24 36    -36 -24 -12  0  12 24 36    -36 -24 -12  0  12 24 36    -36 -24 -12  0  12 24 36

Months since seroconversion

**Fig 2. Individual anti-*Strongyloides* IgG antibody trajectories among 51 participants who presented one or more seroconversions event during consecutive surveys in farming settlements in Amazonas State, Brazil, 2010-2014.** Data are displayed in four panels from left to right and each row within a panel represents an individual (n = 12 in the leftmost panel and **n** = 13 in the other panels). Seroconversion events were defined as an IgG-positive test (absorbance > 0.286) preceded by a negative test 12 months earlier. Samples are ordered in relation to the time of seroconversion and empty spaces indicate missing samples at specific time points. Color codes indicate antibody test results: white means negative (absorbance ≤ 0.286) and coral orange tones indicate quartiles of absorbance in positive samples, 0.287 to 0.366 for the first quartile (lightest tone), 0.367 to 0.561 for the second quartile (coral), 0.562 to 1.121 for the third quartile (darker tone), and above 1.121 for the fourth quartile (brown). By definition, all samples collected 12 months before seroconversion (time point "-12") were negative. Arrows indicate two participants with two seroconversion events each (months 0 and 24).

antibody-positive participants were 5 to 8 times higher than the proportions of people found to pass *S. stercoralis* larvae in the stools in the same communities [25,26]. Importantly, community-wide seroprevance rates in our and other Amazonian populations often exceed the 25% threshold to initiate preventive chemotherapy for strongyloidiasis control, according to the 2024 World Health Organization guideline [27,28]. Median infection prevalence rates ranging between 2.4% and 23.4% were recently estimated for 16 villages in the Amazon region of Ecuador, by combining commercial serology, in-house PCR, and a modified Baermann method [6].

We show that baseline antibody levels in our population are associated with antibody response duration. Indeed, study participants with high antibody levels usually remained seropositive for up to four years, as expected from the ability of adult female worms to persist in the host, due to repeated cycles of autoinfection, and elicit long-lasting antibody responses [1]. We suggest that participants with long-lasting high-level antibody responses may be frequently re-exposed to infective *S. stercoralis* larvae in the soil, may chronically carry adult female worms, or both. Conversely, more than one-third of participants with low anti-*Strongyloides* antibody levels at the baseline became seronegative within 12 months. Low-level and transient IgG responses to larval antigens may be due to exposure to third-stage larvae that failed to develop into adult worms, the spontaneous clearance of infections with adult *S. stercoralis* females, or the presence of cross-reactive antibodies with other STHs. We note, however, that carriage of intestinal nematodes is relatively infrequent in the study population, as judged by the results of 152 fecal-based diagnostic tests carried out during the first survey (S1 Table), most likely due to the frequent self-administration of anthelminthics.

At present, *S. stercoralis* infections are not specifically targeted by STH control programs in the Amazon Basin. The mass administration of albendazole appears to have a negligible impact on the local prevalence of strongyloidiasis [29]. We argue that repeated serosurveys

can provide key information for planning, monitoring, and evaluating control interventions – e.g., those based on ivermectin preventive chemotherapy where the community-wide prevalence of infection diagnosed by stool examination exceeds 10% or the seroprevalence exceeds 25% [27,28] – across the Amazon and other endemic regions in the tropics. Although here we used frozen plasma separated from venous blood samples for serology, small volumes of finger-prick capillary blood spotted onto filter paper are also suitable for antibody testing, with potentially greater acceptability by target populations [6].

The prospective panel design, with repeated antibody measurements in the same participants over time, is a major strength of the present population-based study, as it allowed us to identify temporal trends with potential clinical and public health significance. However, this study has two main limitations. First, sensitive diagnostic tests for strongyloidiasis other than serology were not systematically used for comparison. Limited fecal-based diagnostic data were available at the baseline, but they are likely to have severely underestimated the prevalence of infection. Second, false-positive serology results may arise from past exposure to *S. stercoralis* larvae or adult worms and from cross-reactivity with other STHs [5]. Levels of IgG antibodies may help to distinguish between current vs. past *S. stercoralis* infections. On the one hand, low-level and transient antibody responses were often observed (Fig 2) and many seroconversion events were likely elicited by exposure to third-stage larvae that did not result in intestinal infection. On the other hand, participants with high levels of specific IgG antibodies maintained their seropositive status over extended periods of time (Fig 1), consistent with the chronic carriage of adult worms. Importantly, individuals with long-lasting infections are key contributors to soil contamination and community-wide *S. stercoralis* transmission – therefore, they represent a priority target for control interventions, including (but not limited to) preventive chemotherapy. An additional limitation is the use of plasma samples collected over one decade ago. Because these samples have been stored at -70°C before testing, a substantial decrease in antibody concentrations since collection is not expected. However, this study describes antibody prevalence rates between 2010 and 2014, with no further follow-up. Further studies are needed to explore the current clinical and public health significance of anti-*Strongyloides* antibody carriage and how antibody status varies over time in this and other endemic settings and in response to mass treatment, and to identify serological correlates of chronic infection.

In conclusion, our results further support the use of serology as a diagnostic method that can be implemented in the field, especially as a tool for monitoring temporal changes in *S. stercoralis* transmission intensity in remote populations and estimating the effectiveness of control interventions at a population level.

## Supporting information

**S1 Checklist. STROBE checklist.**
(PDF)

**S1 Fig. Study site. Location of the field site, Remansinho, southern Amazonas State, Brazilian Amazonia.** The map also shows the village of Nova Califórnia (western Rondônia State), the nearest town, Acrelândia (eastern Acre State), where our field laboratory (where blood and stool samples were processed) is situated, and the BR 364 interstate highway, which connects Acre, Rondônia and southern Amazonas to the rest of the country. Source [17]:. The source is an open-access article distributed under the terms of the Creative Commons Attribution License, which permits unrestricted use, distribution, and reproduction in any medium, provided the original author and source are credited.
(PDF)

**S1 Table. Prevalence of infection with intestinal parasites at the baseline in the population of five farming settlements of Amazonas State, Brazil, 2010.**
(PDF)

**S2 Table. Pairwise comparisons of anti-*Strongyloides* IgG status (stratified as negative, low, and high) in consecutive surveys of the population of five farming settlements in Amazonas State, Brazil, with blood draws approximately 12 months apart (2010 vs. 2011, 2011 vs. 2012, 2012 vs 2013, and 2013 vs. 2014).** The last rows provide data for all samples combined.
(DOCX)

**S3 Table. Pairwise comparisons of anti-*Strongyloides* IgG status (stratified as negative, low, and high) in consecutive surveys of the population of five farming settlements in Amazonas State, Brazil, with blood draws approximately 24 months apart (2010 vs. 2012, 2011 vs. 2013, and 2012 vs. 2014).**
(PDF)

**S4 Table. Pairwise comparisons of anti-*Strongyloides* IgG status (stratified as negative, low, and high) in consecutive surveys of the population of five farming settlements in Amazonas State, Brazil, with blood draws approximately 36 months apart (2010 vs. 2013 and 2011 vs. 2014).**
(PDF)

**S5 Table. Pairwise comparisons of anti-*Strongyloides* IgG status (stratified as negative, low, and high) in consecutive surveys of the population of five farming settlements in Amazonas State, Brazil, with blood draws approximately 48 months apart (2010 vs. 2014).**
(PDF)

**S6 Table. Median levels of anti-*Strongyloides* IgG levels (absorbance values) in relation to the time of the seroconversion event (between 2010 and 2013) in the population of five farming settlements in Amazonas State, Brazil.**
(PDF)

**S7 Table. Spearman correlation test results for anti-*Strongyloides* IgG levels (absorbance values) measured in consecutive surveys (between 2010 and 2014) of the population of five farming settlements in Amazonas State, Brazil.**
(PDF)

**S1 Database. Stata file with all variables used in the logistic regression models (DTA format).**
(DTA)

**S2 Database. Stata file with sequential serological data, wide format (DTA format).**
(DTA)

## Acknowledgments

We thank all inhabitants in Remansinho for their enthusiastic participation in this study; the leadership and staff of the Department of Health and Sanitation of Acrelândia for logistic support; Amanda B. Gozze, Susana Barbosa, Nathália F. Lima, Camilla L. Batista, Vanessa C. Nicolete, Pablo S. Fontoura, Susana Ariane S. Viana, Rosely dos Santos Malafronte, and Cristiana F. Alves de Brito for fieldwork; Carla Roberta O. Carvalho and Mauro R. Tucci for

clinical support; Márcio C. Santana, Andrecresa N. Duarte, and Francisco Naildo C. Leitão for overall support during fieldwork; Susana Barbosa for data management; and Anaclara Pincelli for drawing Fig 1.

## Author contributions

**Conceptualization:** Mônica da Silva-Nunes, Carlos E. Cavasini, Kézia K. G. Scopel, Marcelo U. Ferreira.

**Data curation:** Fabiana M. de Paula, Ronaldo C. B. Gryschek.

**Formal analysis:** Fabiana M. de Paula, Ronaldo C. B. Gryschek, Marcelo U. Ferreira.

**Funding acquisition:** Fabiana M. de Paula, Ronaldo C. B. Gryschek, Marcelo U. Ferreira.

**Investigation:** Bruna B. Gomes, Dirce Mary C.L. Meisel, Mônica da Silva-Nunes, Carlos E. Cavasini, Kézia K. G. Scopel, Marcelo U. Ferreira.

**Methodology:** Bruna B. Gomes, Dirce Mary C.L. Meisel.

**Writing – original draft:** Fabiana M. de Paula, Ronaldo C. B. Gryschek, Marcelo U. Ferreira.

**Writing – review & editing:** Fabiana M. de Paula, Bruna B. Gomes, Dirce Mary C.L. Meisel, Mônica da Silva-Nunes, Carlos E. Cavasini, Kézia K. G. Scopel, Ronaldo C. B. Gryschek, Marcelo U. Ferreira.

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
