## [Decision Letter · Decision Letter 0]

5 Feb 2025

PNTD-D-24-01554

Dynamics of anti-Strongyloides IgG antibody responses and implications for strongyloidiasis surveillance in rural Amazonians: a population-based panel data analysis

Dear Dr. Ferreira,

Thank you for submitting your manuscript to PLOS Neglected Tropical Diseases. After careful consideration, we feel that it has merit but does not fully meet PLOS Neglected Tropical Diseases's publication criteria as it currently stands. Therefore, we invite you to submit a revised version of the manuscript that addresses the points raised during the review process.

Please submit your revised manuscript within 60 days Apr 06 2025 11:59PM. If you will need more time than this to complete your revisions, please reply to this message or contact the journal office at plosntds@plos.org. Please include the following items when submitting your revised manuscript:

We look forward to receiving your revised manuscript.

Kind regards,

Alessandra Morassutti, PhD

Academic Editor

Krystyna Cwiklinski

Section Editor

Shaden Kamhawi

co-Editor-in-Chief

Paul Brindley

co-Editor-in-Chief

**Additional Editor Comments (if provided):**

Dear authors, the reviewers of this article have pointed out some issues that need to be addressed, especially in relation to methodology. Please address all coments, even if the authors disagree with any of the questions or suggestions.

**Journal Requirements:**

1) Tables should not be uploaded as individual files. Please remove these files and include the Tables in your manuscript file as editable, cell-based objects. For more information about how to format tables, see our guidelines:

https://journals.plos.org/plosntds/s/tables

2) Some material included in your submission may be copyrighted. According to PLOSu2019s copyright policy, authors who use figures or other material (e.g., graphics, clipart, maps) from another author or copyright holder must demonstrate or obtain permission to publish this material under the Creative Commons Attribution 4.0 International (CC BY 4.0) License used by PLOS journals. Please closely review the details of PLOSu2019s copyright requirements here: PLOS Licenses and Copyright. If you need to request permissions from a copyright holder, you may use PLOS's Copyright Content Permission form.

Potential Copyright Issues:

- Figure S1. Please provide a direct link to the base layer of the map (i.e., the country or region border shape) and ensure this is also included in the figure legend; and provide a link to the terms of use / license information for the base layer image or shapefile. We cannot publish proprietary or copyrighted maps (e.g. Google Maps, Mapquest) and the terms of use for your map base layer must be compatible with our CC BY 4.0 license.

3) Please ensure that the funders and grant numbers match between the Financial Disclosure field and the Funding Information tab in your submission form. Note that the funders must be provided in the same order in both places as well. State the initials, alongside each funding source, of each author to receive each grant. For example: "This work was supported by the National Institutes of Health (####### to AM; ###### to CJ) and the National Science Foundation (###### to AM).".

**Reviewers' Comments:**

Reviewer's Responses to Questions

**Key Review Criteria Required for Acceptance?**

**Methods** :

-Are the objectives of the study clearly articulated with a clear testable hypothesis stated?

-Is the study design appropriate to address the stated objectives?

-Is the population clearly described and appropriate for the hypothesis being tested?

-Is the sample size sufficient to ensure adequate power to address the hypothesis being tested?

-Were correct statistical analysis used to support conclusions?

-Are there concerns about ethical or regulatory requirements being met?

Reviewer #1: Please see below.

Reviewer #2: The laboratory procedures are well described.

The statistical analysis is mostly well described, but see my specific feedback under general comments with regards to the description of how the logistic regression model was built.

Although the ethics statement is thorough, it doesn't mention the original purpose of sampling (presumably for a malaria study) and any relevance of this.

**Results** :

-Does the analysis presented match the analysis plan?

-Are the results clearly and completely presented?

-Are the figures (Tables, Images) of sufficient quality for clarity?

Reviewer #1: Please see below.

Reviewer #2: The analysis presented broadly matches the analysis plan, although I think the order of the results section could be revised (starting with clarification over which samples were tested, and for what - serology, faecal etc, with timings of surveys).

As per my specific feedback in the summary and general comments below, the tables and figures need some revision to enhance clarity.

**Conclusions** :

-Are the conclusions supported by the data presented?

-Are the limitations of analysis clearly described?

-Do the authors discuss how these data can be helpful to advance our understanding of the topic under study?

-Is public health relevance addressed?

Reviewer #1: Please see below.

Reviewer #2: Limitations – the samples were collected over ten years ago, this should be discussed.

Conclusion – Do you really mean a diagnostic method for clinical purposes (i.e. treating on the basis of positive serology)? Or do you recommend the method more for epidemiological surveillance and to aid decision making on control programmes?

**Editorial and Data Presentation Modifications?**

Reviewer #1: Figure 1 is somewhat confusing and the authors should find a way to include the numbers of 'converters' into their figure.

Reviewer #2: As per my summary and general comments below, the tables and figures need some revision to enhance clarity.

**Summary and General Comments** :

Reviewer #1: Review on PNTD-D-24-02401554 by de Paula and colleagues entitled ‘Dynamics of anti-Strongyloides IgG antibody responses and implications for strongyloidiasis surveillance in rural Amazonians: a population-based panel data analysis’.

Comments:

In the present study, the authors applied repeated antibody testing to estimate the prevalence of anti-Strongyloides IgG responses and identify incident infections in five farming settlements in the Amazonas State of Brazil. In 2010, the authors performed a longitudinal, serological study in different rural settlements. However, for comparison, only one fecal examination was performed in 2010 in enrolled individuals, without further parasitological or molecular testing.

The manuscript has an adequate title and is written in sound English language.

Major Comments:

-Material and Methods, lines 165ff.: How was the transport chain and continuous freezing guaranteed for the sensitive biological samples from the remote Amazon area? Please comment on shipment and storage of samples. From the description, it is not clear whether the immunoassays were carried out soon after collection, or whether they were carried out only most recently. Please explain.

-Materials and Methods, study population: The authors stated that only patients with microscopically confirmed infections were treated with anti-parasitic drugs. What happened with the serologically positive individuals with antibodies against Strongyloides and especially with those who seroconverted? Is there any federal guideline from the Brazilian Ministry of Health on how to deal with those individuals?

-Results, Lines 278-287: Since the presence of other intestinal helminth infections is important for the interpretation of positive serology for strongyloidiasis, the authors should give a brief description of parasitological results here, at least in terms of presence (n; %) of other intestinal helminth infections. Why parasitological exams were discontinued in the following years?

-Results, Lines 338-340 and 345-348: The authors should not include and discuss speculative comments into the Results section.

-Results: My major concern is the absence of parasitological exams in follow up visits and the very speculative discussion on Strongyloides-derived seroconversion during the longitudinal study, without a strong, parasitological fundament (Baermann, APC or PCR). Additional parasitological exams (e.g. K-K or HPJ), even not adequate for the diagnosis of Strongyloides, would, at least, be useful to exclude the presence of other soil-transmitted helminth infections. As stated before by the authors, this might facilitate the participation of other helminth infections and the occurrence of cross-reactivities, which should not be attributed to the presence of Strongyloides. This weak point is critically addressed and discussed by the authors in the manuscript, however, does not reduce my concerns in terms of validity of the results for the epidemiology of strongyloidiasis, especially in this remote Amazon area, where other intestinal infection are supposed to be frequent. A possibility for reducing the cross-reactivity with hookworm or Ascaris-derived antigens would be the preabsorption of plasma samples with these antigens. Did the authors consider this?

-Discussion: The authors might include some discussion on how these seropositive individuals should be dealt with and whether MDA campaigns with ivermection should be considered. This might be important for the scenario of anti-Strongyloides campaigns in Brazil and elsewhere.

Minor Points:

-Materials and Methods, Line 214: Please use ‘higher’ instead of ‘greater’.

-Results, Lines 323-324: The Results section should not contain a discussion part and indications of references from other research groups [ref 22 and 23]. This is very speculative here, since no further fecal examinations were conducted.

Reviewer #2: This paper addresses an important topic and my overall impression is that it is mostly well-written, with a thorough description of the laboratory procedures. Some clarifications to the manuscript are required to better orient the reader, to improve the clarity in presentation of the results, and to discuss the public health implications of the findings. I suggest the following issues to be addressed:

1. Title – is dynamics really the right word?

2. Line 94 [ref. 5; but see also ref 6] – it’s not clear why you’ve chosen to cite the references this way. Please clarify why you’re not just referencing both [5] and [6].

3. The low sensitivity of faecal-based diagnostics is well described, but can you clarify the issue with diagnostics further. Is the low sensitivity more of a problem clinically, or just for epidemiological surveillance?

4. Line 400. Cross-reactivity with other locally prevalent STHs is mentioned but can you be more specific as this is a vital component to understand in order to interpret your findings? Either expand on this here, or perhaps it is better suited in the discussion.

5. Line 116. One-fourth – change to a quarter or 25%

6. Line 137 onwards. It’s very helpful for readers to have more background information about the setting, e.g. healthcare access, broader health status. Why were these communities chosen and how representative are they – are the results applicable to other populations?

7. Ethics statement. This is a thorough description (probably not all needed for the manuscript), however doesn’t mention the purpose of the original study. Am I correct in thinking the samples were originally taken (and participants consented) to a study about malaria? Did participants consent for future research but not explicitly for a study about strongyloidiasis?

8. Line 166. If normally distribute please present mean and standard deviation. If not, median and interquartile range.

9. Line 168. ‘Because of the high mobility of the local population, only 5.4% (n=24) of the study participants had specific antibodies measure in study years’. I don’t understand what you mean here. Please clarify.

10. It’s not very clear from the methods how many participants had which type of samples and tests, and the sequencing of this. Perhaps a table in the results section – with rows depicting samples and test and columns dates of surveys, would be helpful to orientate the reader.

11. Please clarify – was it just participants with faecal diagnosis of strongyloidiasis that were offered treatment with ivermectin? Not those with positive serology?

12. Line 209 – ‘with confirmed S. stercoralis infection’ – how was it confirmed? Please make it clear throughout what you took to be the gold-standard confirmation of diagnosis.

13. Line 227 onwards. Description of variables in logistic regression model well described, but how was the model built? Did you put all the variables in together and only keep the statistically significant ones, or did you sequentially add them one by one. If the latter, what order did you add them?

14. Line 253 – how were confidence intervals estimated?

15. Results section – as above I think a table denoting what samples and tests were done (with timings) would be helpful at the beginning of the results section.

16. Line 263 – can you present how many individuals ever had positive serology first, then how many samples in total?

17. Table 1 needs some clarification. The n/N(%) column needs a clearer explanation e.g. Positive S. stercoralis ELISA/number of samples tested (seroprevalence, %). I don’t think it’s enough to only have the explanation in the ledger.

18. Table 1. Spelling of antihelminthics needs correcting in table 1.

19. Table 1. Use of antihelminthics and passed helminths – it needs to be clear that this is self-reported from a survey (if that’s correct).

20. Line 273 – although increase in antibody prevalence noted, the confidence intervals overlap.

21. Line 281 – ‘periodic deworming was a common practice’. Can you be more specific about this (expanding on it in the discussion section would be helpful).

22. Line 282 – one fourth – change to a quarter or give exact percentage.

23. Figure 1. This is helpful, but is it missing the ledger to explain what A, B, C & D refer to?

24. Figure 2. Again needs a ledger to explain the colour grading. All tables and figures should be revised to make sense alone without having to refer to the results section text for interpretation.

25. Line 323 – exactly how many had reported recent deworming with albendazole or mebendazole? This is important as could potentially explain seroreversion.

26. Line 332 – please correct the typo. Samples -> sample.

27. Discussion section: Can you add a very brief summary of the key results at the beginning of the discussion? I think it would be clearer if you started the discussion with the paragraph beginning line 371 (and paragraph line 377) – then put these into context with the prevalence estimates from the literature (line 361).

28. Limitations – the samples were collected over ten years ago, this should be discussed.

29. Conclusion – Do you really mean a diagnostic method for clinical purposes (i.e. treating on the basis of positive serology)? Or do you recommend the method more for epidemiological surveillance and to aid decision making on control programmes?

PLOS authors have the option to publish the peer review history of their article (what does this mean? ). If published, this will include your full peer review and any attached files.

**Do you want your identity to be public for this peer review?** For information about this choice, including consent withdrawal, please see our Privacy Policy .

Reviewer #1: No

Reviewer #2: No

**Figure resubmission:**
---

## [Decision Letter · Decision Letter 1]

6 Mar 2025

Dear Dr Ferreira,We are pleased to inform you that your manuscript 'Dynamics of anti-Strongyloides IgG antibody responses and implications for strongyloidiasis surveillance in rural Amazonians: a population-based panel data analysis' has been provisionally accepted for publication in PLOS Neglected Tropical Diseases.

Best regards,

Alessandra Morassutti, PhD

Academic Editor

Krystyna Cwiklinski

Section Editor

Shaden Kamhawi

co-Editor-in-Chief

Paul Brindley

co-Editor-in-Chief

Reviewer's Responses to Questions

**Key Review Criteria Required for Acceptance?**

**Methods**

-Are the objectives of the study clearly articulated with a clear testable hypothesis stated?

-Is the study design appropriate to address the stated objectives?

-Is the population clearly described and appropriate for the hypothesis being tested?

-Is the sample size sufficient to ensure adequate power to address the hypothesis being tested?

-Were correct statistical analysis used to support conclusions?

-Are there concerns about ethical or regulatory requirements being met?

Reviewer #1: No further comments.

Reviewer #2: Thank you for adequately addressing all the points raised in the feedback

**Results**

-Does the analysis presented match the analysis plan?

-Are the results clearly and completely presented?

-Are the figures (Tables, Images) of sufficient quality for clarity?

Reviewer #1: No further comments.

Reviewer #2: Thank you for adequately addressing all the points raised in the feedback

**Conclusions**

-Are the conclusions supported by the data presented?

-Are the limitations of analysis clearly described?

-Do the authors discuss how these data can be helpful to advance our understanding of the topic under study?

-Is public health relevance addressed?

Reviewer #1: No further comments.

Reviewer #2: Thank you for adequately addressing all the points raised in the feedback

**Editorial and Data Presentation Modifications?**

Reviewer #1: No further comments.

Reviewer #2: The modifications from the first review have enhanced the clarity of the manuscript, I recommend to accept.

**Summary and General Comments**

Reviewer #1: The authors have attended all issues raised by the reviewer. As such, the manuscript might be accepted.

Reviewer #2: This paper addresses an important topic and my overall impression is that it is mostly well-written, with a thorough description of the laboratory procedures. The authors have provided some important points of clarification which have significantly improved the readability of the manuscript.

PLOS authors have the option to publish the peer review history of their article (what does this mean? ). If published, this will include your full peer review and any attached files.

**Do you want your identity to be public for this peer review?** For information about this choice, including consent withdrawal, please see our Privacy Policy .

Reviewer #1: No

Reviewer #2: No

---

## [Editor Report · Acceptance letter]

Dear Dr. Ferreira,

We are delighted to inform you that your manuscript, "Dynamics of anti-Strongyloides IgG antibody responses and implications for strongyloidiasis surveillance in rural Amazonians: a population-based panel data analysis," has been formally accepted for publication in PLOS Neglected Tropical Diseases.

Best regards,

Shaden Kamhawi

co-Editor-in-Chief

Paul Brindley

co-Editor-in-Chief
